# Synergy of Human Platelet-Derived Extracellular Vesicles with Secretome Proteins Promotes Regenerative Functions

**DOI:** 10.3390/biomedicines10020238

**Published:** 2022-01-23

**Authors:** Fausto Gueths Gomes, André Cronemberger Andrade, Martin Wolf, Sarah Hochmann, Linda Krisch, Nicole Maeding, Christof Regl, Rodolphe Poupardin, Patricia Ebner-Peking, Christian G. Huber, Nicole Meisner-Kober, Katharina Schallmoser, Dirk Strunk

**Affiliations:** 1Cell Therapy Institute, Spinal Cord Injury and Tissue Regeneration Center Salzburg (SCI-TReCS), Paracelsus Medical University (PMU), 5020 Salzburg, Austria; fausto.gueths@pmu.ac.at (F.G.G.); andre.cronemberger@pmu.ac.at (A.C.A.); martin.wolf@pmu.ac.at (M.W.); sarah.hochmann@pmu.ac.at (S.H.); linda.krisch@pmu.ac.at (L.K.); nicole.maeding@pmu.ac.at (N.M.); rodolphe.poupardin@pmu.ac.at (R.P.); patricia.ebner@pmu.ac.at (P.E.-P.); 2Department of Transfusion Medicine and SCI-TReCS, Paracelsus Medical University (PMU), 5020 Salzburg, Austria; k.schallmoser@salk.at; 3Department for Biosciences and Medical Biology, Paris Lodron University, 5020 Salzburg, Austria; christof.regl@plus.ac.at (C.R.); c.huber@plus.ac.at (C.G.H.); nicole.meisner-kober@plus.ac.at (N.M.-K.)

**Keywords:** human platelet lysate (HPL), platelet-rich plasma (PRP), extracellular vesicles (EVs), corona, skin regeneration, immune modulation, angiogenesis

## Abstract

Platelet-rich plasma is a promising regenerative therapeutic with controversial efficacy. We and others have previously demonstrated regenerative functions of human platelet lysate (HPL) as an alternative platelet-derived product. Here we separated extracellular vesicles (EVs) from soluble factors of HPL to understand the mode of action during skin-organoid formation and immune modulation as model systems for tissue regeneration. HPL-EVs were isolated by tangential-flow filtration (TFF) and further purified by size-exclusion chromatography (SEC) separating EVs from (lipo)protein-enriched soluble fractions. We characterized samples by tunable resistive pulse sensing, western blot, tandem mass-tag proteomics and super-resolution microscopy. We evaluated EV function during angiogenesis, wound healing, organoid formation and immune modulation. We characterized EV enrichment by TFF and SEC according to MISEV2018 guidelines. Proteomics showed three major clusters of protein composition separating TSEC-EVs from HPL clustering with TFF soluble fractions and TFF-EVs clustering with TSEC soluble fractions, respectively. HPL-derived TFF-EVs promoted skin-organoid formation and inhibited T-cell proliferation more efficiently than TSEC-EVs or TSEC-soluble fractions. Recombining TSEC-EVs with TSEC soluble fractions re-capitulated TFF-EV effects. Zeta potential and super-resolution imaging further evidenced protein corona formation on TFF-EVs. Corona depletion on SEC-EVs could be artificially reconstituted by TSEC late fraction add-back. In contrast to synthetic nanoparticles, which commonly experience reduced function after corona formation, the corona-bearing EVs displayed improved functionality. We conclude that permissive isolation technology, such as TFF, and better understanding of the mechanism of EV corona function are required to realize the complete potential of platelet-based regenerative therapies.

## 1. Introduction

Platelets are key players in primary hemostasis preventing blood loss at sites of vascular injury [1], but also in innate and adaptive immune defense [2]. They support wound healing by secreting growth factors, such as platelet-derived growth factor β (PDGF-β), or by stimulating resident stromal cells to secrete cytokines inducing wound healing and tissue regeneration [3,4,5]. Despite initial enthusiasm, the broad application of platelet-derived products in regenerative medicine is hampered by a lack of definitive evidence [6,7]. In particular, platelet-rich plasma (PRP) treatment did not facilitate significant improvement of musculoskeletal or soft tissue injuries, as demonstrated by a recent Cochrane Foundation meta-analysis [8]. Several groups are therefore evaluating the applicability of novel platelet derivatives, including human platelet lysate (HPL), for regenerative purposes [9,10]. HPL attracted particular attention as a promising raw material for cell therapy manufacture and it can be prepared cost-effectively via lysis of outdated clinical platelet concentrates [11,12]. We have recently demonstrated that HPL can also significantly accelerate human skin-organoid formation in vitro and wound healing in vivo [13]. HPL may thus be an interesting platelet-derived biomaterial for regenerative medicine applications, acting via release of soluble growth factors stored within platelets [5] or via extracellular vesicles (EVs) [14].

Platelet EVs were first described in a seminal paper by Wolf and co-workers as ‘platelet dust’ in 1967 [15]. With time, they were recognized to be the most frequent population of EVs in healthy human plasma [16]. Depending on how platelets are activated, their secreted platelet EV cargo differs [17]. Functionally, platelet EVs have been described to promote clot formation, angiogenesis and wound healing [18]. The precise molecular characterization of EV functions has been complicated because additional biological nanoparticles such as lipoproteins or protein aggregates co-purify with EVs from biological fluids [19]. Among biological nanoparticles in blood, the frequency of lipoproteins was estimated to be more than one million-fold higher than that of platelet EVs [20].

Commonalities and differences exist between biological and synthetic nanoparticles concerning size, physical characteristics and cargo delivery to target cells [20,21]. A key observation during preclinical testing of synthetic nanoparticles was that they acquired a protein corona coating their surface particularly during contact with plasma, frequently resulting in a loss of function [22]. Extensive nanotechnology research revealed that the nanoparticle corona can be divided into an inner layer hard corona of strong physico-chemical interactions adjacent to the particle’s surface and a so-called soft corona of weaker interactions forming an outer layer [23]. We and others discovered recently, that EVs from different sources can also bear a biologically active protein corona [24,25]. Its composition could be altered depending on the protein milieu in which the EVs are present [25] and by the method of separation [24].

Here we asked whether HPL-derived EVs or the platelet-derived soluble factors mediate the trophic HPL effects. We observed that protein-rich TFF-EVs or a combination of the protein-depleted TSEC-EVs with add-back of protein-rich TSEC-soluble fractions were efficient in four surrogate readouts for regenerative function. The EV-depleted TFF soluble fractions and the protein-depleted TSEC-EVs were not active in these assays. We found a shift in electrical charge in TSEC-EVs in a protein-depleted environment indicating corona formation and we succeeded to artificially reconstitute the protein-corona. These results may point to the interchangeable nature of the EV-corona. We conclude that not only the starting material, but also the isolation/enrichment method(s) of choice can result in EV preparations with different protein corona and functional activity.

## 2. Materials and Methods

### 2.1. HPL Preparation

Pooled HPL was prepared as described [12]. In brief, after written informed consent about the use of residual material, 10 expired buffy-coat derived platelet concentrates from a total of 40 healthy blood donors were frozen at −30 °C overnight. For platelet lysis, the bags were thawed (37 °C) and further pooled into one bag (Macopharma, Mouvaux, France), gently mixed, re-aliquoted into separate bags (Macopharma) and frozen for a second time at −30 °C. The lysates were thawed (37 °C) and centrifuged (4,000× *g*, 16 min, room temperature, RT) to deplete the platelet fragments. The supernatant was collected as pooled HPL. Samples were aliquoted into 50 mL tubes and frozen at ≤ −20 °C for further use.

### 2.2. Isolation of HPL EVs

Twenty HPL aliquots (in total 1 L) were thawed and added to 500 mL bottles (100 mL HPL per bottle) of minimum essential medium Eagle with alpha-modifications (α-MEM, M4526, Sigma-Aldrich, Saint Louis, MO, USA) or RPMI-1640 (R0883, Sigma-Aldrich), supplemented with 200 mg/L CaCl_2_ (Sigma-Aldrich), incubated at 37 °C for 1 h and kept at 4 °C overnight. The next day, gelatinous clots were destructed by vigorous shaking and samples were centrifuged (3000× *g* for 10 min, RT) as previously described [26]. Supernatants were diluted 1:2 with fresh medium, 0.22 µm filtered and subjected to tangential-flow filtration (TFF, KrosFlo KR2i; Repligen, Waltham, MA, USA) with a 500 kDa cut-off as described [24]. The resulting TFF-EVs were collected from the retentate and TFF-separated soluble factors (sol.F.) from the permeate. For further purification, 4 mL of TFF-EVs were separated with a qEV_2_ size-exclusion chromatography (SEC) column (Izon Science, Christ Church, New Zealand), taking a total of 25 fractions of 2 mL each as described [27]. We loaded 4 mL of TFF-EV samples onto SEC columns, taking 2 mL fractions, with a total volume of 6 mL for the TSEC-EV and corresponding soluble-fraction pools (3 fractions each); 25 fractions (50 mL) total volume were obtained after SEC. TSEC-EVs were eluted on fractions 7–9 and pooled, while protein-rich soluble fractions (TSEC-soluble fractions) were pooled from fractions 19–21. Both TFF and SEC were performed using 0.9% saline containing 10 mM HEPES (Sigma-Aldrich) as buffer. Graphical depiction of the protocol above is shown in the results section.

### 2.3. EV Quantification by Tunable Resistive Pulse Sensing (TRPS)

EVs were characterized according to concentration, size and zeta potential using a qNano Gold analyzer (Izon Science) equipped with a NP150 nanopore, optimal for measurement of particles sized 70–450 nm, as described [24]. For concentration and size, 10 Pa pressure was applied onto the samples and calibration beads (Izon Science) in combination with 130–140 nA electrical current.

### 2.4. Protein Quantification and Western Blot

Sample protein concentration was determined using detergent-compatible (DC) protein assay (Bio-Rad, Hercules, CA, USA) according to the manufacturer’s instructions. Optical density (OD) at 750 nm was measured with a SPARK 7 plate reader (Tecan, Groedig, Austria). Western blot samples were loaded onto a pre-casted SDS-page gel (4–20%; Bio-Rad) in Laemmli buffer under non-reducing conditions as described [28], or tetraspanins CD9 and CD63, and containing 50 µM dithiothreitol (DTT) as reducing agent for all other targets, together with a molecular weight marker (Precision Plus protein dual Xtra prestained protein standard, Bio-Rad) [29]. In brief, runs were made at 20 mA per gel, 1 h, in Tris/Glycine/SDS buffer on a protean mini system (Bio-Rad). Samples were transferred to a wet system, on ice, 120 V for 1.5 h to polyvinylidene difluoride (PVDF) membranes (Bio-Rad), blocked for 30 min (shaking) on 2% bovine serum albumin (BSA; Sigma-Aldrich) or 5% skim-milk (only for human-serum albumin; Bio-Rad). Primary antibodies were added in 1% BSA-containing tris-buffered saline (TBS) with 0.05% tween (TBST; or 2% milk plus TBST for anti-human serum albumin antibody) overnight (4 °C, shaking). Primary antibodies were CD9 (0.05 µg/mL, clone MM2/57, Life Technologies, Carlsbad, CA, USA), CD63 (0.05 µg/mL, clone TS63, Thermo Fisher Scientific, Waltham, MA, USA), apolipoprotein A1 (ApoA1, 0.132 µg/mL, GTX112692, GeneTex, Irvine, CA, USA) and human serum albumin (HSA, KT11, 0.1 µg/mL, Invitrogen, Waltham, MA, USA). Enhanced chemiluminescence (ECL) clarity substrate (Bio-Rad) was added to the membranes and bands were visualized on a ChemiDoc^TM^ imaging system and Image Lab software (both Bio-Rad).

### 2.5. Tandem Mass-Tag (TMT) Proteomics

#### 2.5.1. Chemicals

All solutions were prepared with ultrapure water produced in-house with a MilliQ^®^ Integral 3 instrument (Millipore, Billerica, MA, USA). Methanol (MeOH, ≥99.9%) and acetonitrile (ACN, ≥99.9%) were acquired from VWR International (Vienna, Austria). Triethylammonium bicarbonate buffer (TEAB, pH 8.5 ± 0.1, 1 mol/L), sodium dodecyl sulfate (SDS, ≥99.5%), tris(2-carboxyethyl)phosphin-hydrochlorid (TCEP, ≥98.0%), iodoacetamide (IAA, ≥99.0%), formic acid–(FA, 98–100%) and trifluoroacetic acid (TFA ≥ 99.0%) were purchased from Sigma-Aldrich (Vienna, Austria). Ortho-phosphoric acid (85%) was obtained from Merck (Darmstadt, Germany). Trypsin (sequencing grade modified, porcine) was purchased from Promega (Madison, WI, USA).

#### 2.5.2. Sample Preparation

Samples were prepared in 5% SDS in 50 mmol/L TEAB (pH 8.5) at 95 °C for 5 min followed by sonication (Bioruptor, Diagenode, Liège, Belgium) for 10 min. After a 10 sec centrifugation step at 13,000× *g*, protein content in the supernatant was analyzed by a Pierce BCA Protein assay kit (ThermoFisher Scientific, Vienna, Austria) according to the manufacturer’s instructions. Next, 50 µg protein per sample was treated with 5 mmol/L TCEP at 55 °C for 15 min to reduce disulfides, followed by alkylation of the cysteine residues by addition of IAA to a concentration of 40 mmol/L, and incubation at 22 °C in the dark for 15 min. Subsequently, the samples were acidified to pH ≤ 1 with ortho-phosphoric acid before purification employing S-Trap micro columns (Protifi, Huntington, NY, USA) according to the manufacturer’s instructions. For proteolysis, trypsin was added at an enzyme to protein ration of 1:20 (*w*/*w*) followed by overnight incubation at 37 °C. The obtained peptides were eluted from the S-trap columns, dried at 40 °C in a vacuum centrifuge, resuspended in 100 mmol/L TEAB (pH 8.5) and labeled by TMTpro™ 16plex (A44521, WF324547, ThermoFisher Scientific, Vienna, Austria) following the manufacturer’s protocol. A pool of the three biological replicates was labelled with the 126C channel to serve as reference channel.

#### 2.5.3. High-Performance Liquid Chromatography Coupled to Mass Spectrometry

Chromatographic separation of 4 µg of sample was carried out on an UltiMate™ 3000 RSLCnano System (ThermoFisher Scientific, Germering, Germany), employing reversed phase HPLC using a µPAC™ C18 trapping column (10 × 2 mm i.d., 0.25 µL total volume) and a µPAC™ C18 separation column (2000 × 0.04 mm i.d.) both from PharmaFluidics, Ghent, Belgium. To trap the peptides, 0.1% aqueous TFA 1% ACN was used as mobile phase at a flow rate of 2 µL/min for 10 min. For the separation, 0.1% aqueous FA (solvent A) and 0.1% FA in ACN (solvent B) were pumped at a flow rate of 300 nL/min in the following order: 1% B for 5 min, a linear gradient from 1–3% B in 10 min, a second linear gradient from 3–21% B in 350 min, and a third linear gradient from 21–40% B in 165 min. This was followed by flushing at 80% B for 15 min and column re-equilibration at 1% B for 40 min. The column temperature was kept constant at 50 °C. The nanoHPLC system was hyphenated to a Q Exactive™ Plus Hybrid Quadrupole-Orbitrap™ mass spectrometer via a Nanospray Flex™ ion source (both from Thermo Fisher Scientific, Bremen, Germany). The source was equipped with a SilicaTip emitter with 360 µm o.d., 20 µm i.d. and a tip i.d. of 10 µm purchased from CoAnn Technologies Inc. (Richland, WA, USA). The spray voltage was set to 1.5 kV, S-lens RF level to 55.0 and capillary temperature to 320 °C. Each scan cycle consisted of a full scan at a scan range of *m*/*z* 350–2000 and a resolution setting of 70,000 at *m*/*z* 200, followed by 15 data-dependent higher-energy collisional dissociation (HCD) scans in a 1.2 *m*/*z* isolation window at 30% normalized collision energy at a resolution setting of 35,000 at *m*/*z* 200. For the full scan, the automatic gain control (AGC) target was set to 3e6 charges with a maximum injection time of 150 ms, for the HCD scans the AGC target was 2e5 charges with a maximum injection time of 150 ms. Already fragmented precursor ions were excluded for 30 s. The sample was measured in three technical replicates. Data acquisition was conducted using Thermo Scientific™ Chromeleon™ 7.2 CDS (ThermoFisher Scientific, Germering, Germany).

#### 2.5.4. Data Evaluation

For data evaluation MaxQuant 2.0.1.0 [30] was used in default settings for reporter ion MS2 with weighted ratio to reference channel for normalization and correcting for isotope impurities in the TMTpro reagents. For protein identification a database from the Uniprot consortium [31] including only reviewed Swiss-Prot entries for Homo sapiens (Human, from 25 August 2021) was used applying a 1% false discovery rate. The obtained protein groups were processed using the Perseus software platform [32]. First the protein groups were filtered removing proteins that were only identified by site and reverse sequence matches. Next, the reporter ion intensities were log2 transformed and normalized by subtraction of the median. Finally, Kmeans hierarchical (complexheatmap package) clustering and biplot principal component analysis of normalized z-scores were performed in RStudio (Integrated Development for R, Boston, MA, USA), with additional clusterProfiler analysis [33]. Enrichment analyses of the quantified proteins regarding canonical pathways, were per-formed with QIAGEN IPA (QIAGEN Inc., Hilden, Germany) [34].

### 2.6. Skin-Organoids, Fibrosphere Formation and Immunohistochemistry

Human primary keratinocytes (KCs) and fibroblasts (FBs) were obtained from expanded human skin grafts, cultivated in CntPrime medium (CnT-PR, CELLnTEC, Bern, Switzerland) and α-MEM (Sigma-Aldrich) supplemented with 10% HPL, respectively, as described [13]. Endothelial colony-forming cells (ECFCs) were obtained from human umbilical cord blood as published [35] and cultured in EGM basal medium (CC-3156, Lonza Group, Basel, Switzerland) supplemented with the bullet kit (EGM-2 medium, CC4176, Lonza) with 10% HPL instead of fetal bovine serum (FBS) including 2 IU/mL heparin to produce endothelial cells (ECs) [36]. Fibrosphere formation was induced by adding 5,000 FBs per well in a 96-well ultra-low attachment plate (Nunclon Sphera, ThermoFisher Scientific) and skin-organoids were created by mixing KCs, FBs and ECFCs at a 2:1:1 ratio, respectively, totaling 5000 cells/well for quantification or 50,000 cells/well for immunohistochemistry [13]. Plates were kept at 37 °C, 5% CO_2_, 5% O_2_ for 6 days, followed by imaging on an Eclipse *Ti* microscope (Nikon, Tokyo, Japan), 10x objective, 7 × 8 pictures per well to cover the whole area. Fibrospheres and skin-organoids were quantified using NIS software (Nikon). Samples were aspirated from the wells, centrifuged at 300× *g*, 5 min, and resuspended in 100 µL of human blood group AB plasma (Siemens Healthineers, Erlangen, Germany) plus 100 µL of human thromboplastin (Thromborel^®^ S, Siemens Healthineers), to induce clotting. Clots were transferred to plastic adapters, fixed in 4% paraformaldehyde (PFA), embedded in paraffin and sectioned in 4 µm slices. Mayer’s Hemalaun (#1.09249.2500, Merck Millipore, Burlington, MA, USA) and eosin (#1.15935.0100, Merck Millipore) were used for hematoxylin and eosin (H and E) staining. Immunohistochemistry was performed with anti-human vimentin (1.56 μg/mL, clone V9, M072501-2, Dako, Germany), cytokeratin 14 antibody (40 ng/μL, clone LL001, Santa Cruz, Dallas, TX, USA), secondary antibodies goat anti mouse IgG, Alexa Fluor 488 and 594 (A-11001, A-11005; Invitrogen) and 4′,6-diamidino-2-phenylindole (DAPI mounting media, Agilent).

### 2.7. Immunomodulation Assay

Immunomodulation was assessed by inhibition of T-cell proliferation, as previously shown [37,38]. Briefly, peripheral mononuclear cells (PBMCs, collected from 10 donors after informed written consent) were pooled, pre-stained with 2 µM carboxyfluorescein succinimidylester (CSFE; Sigma-Aldrich) and cryopreserved in multiple aliquots at −182 °C until use. Thawed pre-labeled PBMCs were stimulated with either phytohemagglutinin (PHA; 5 µg/mL; Sigma-Aldrich) or CD3/28 dynabeads (2.4 × 10^4^ beads per well; Thermo Fisher Scientific) in the absence or presence of EV samples as indicated at ratios of 15,000, 5000 and 1700 EVs/cell, in technical multiplicates as indicated, with a total of 300,000 cells/well. TSEC-soluble fraction samples were protein-normalized to the TFF protein load. PBMCs and HPL or EV samples were incubated for 4 days, 37 °C, 5% CO_2_, 18% O_2_. Samples were stained with anti-CD3-APC antibody (2.5 µg/mL, clone SK7, BD), for 30 min at 4 °C, and the viability dye 7-AAD (1 µg/mL; Thermo Fisher). A minimum of 10,000 events gated on viable CD3^+^ cells were acquired on a Gallios flow cytometer (Beckman Coulter, Brea, CA, USA) and analyzed with Kaluza software 1.3 (Beckman Coulter). Proliferating T cells were gated at CFSE^low^. PHA- or CD3/28-untreated proliferating cells were normalized as 0% inhibition of T cell proliferation, and relative inhibition was then assessed on sample-treated conditions.

### 2.8. Zeta Potential and Size Measurements by Nanoparticle Tracking Analysis (NTA)

NTA was performed on a ZetaView PMC 110 (Particle Metrix GmbH, Inning, Germany) in light scatter mode, as described [39]. Briefly, samples were diluted in Dulbecco’s phosphate buffered saline (DPBS; Sigma-Aldrich), and 1 mL was applied into the measurement cell. Pulsed zeta potential and size were measured in 11 different fields per replicate.

### 2.9. Super-Resolution Microscopy

Direct stochastic optical reconstruction microscopy (dSTORM) was performed with a Nanoimager S (Oxford Nanoimaging; ONI; Oxford, UK). HPL-derived TSEC-EVs (5.4 × 10^8^) were pre-incubated with 3 µg of fluorescent bovine serum albumin-Alexa Fluor (AF)-488 (Invitrogen), for 30 min RT in the dark, and then stained overnight at 4 °C with a tetramix of anti-CD9 (20 µg/mL; FAB1880R; R&D Systems, Minneapolis, MN, USA), anti-CD63 (50 µg/mL; 561983; BD) and anti-CD81 (20 µg/mL; tetramix, FAB4615R; R&D), all directly conjugated to AF-647, and loaded thereafter onto EV profiler chips (ONI), designed to bind EVs specifically, according to the manufacturer’s instructions. Images were acquired at 30% (51.7 mW) and 50% (66.4 mW) laser power for 647 nm and 488 nm, respectively, 2500 frames each. Images were analyzed using the collaborative discovery (CODI) online analysis platform employing drift at minimum entropy (DME) correction, 16–4000 localizations, 10–1000 nm radius and maximum 180 nm distance in signals to be considered co-localized [28].

### 2.10. Angiogenesis Assay

In vitro angiogenesis assay was performed as described previously [24,28]. Briefly, endothelial cells (ECs) were starved for 18 h before seeding 6000 cells/well in an angiogenesis 96-well µ-plate (Ibidi, Gräfelfing, Germany) pre-coated with reduced growth factor basement membrane matrix (Geltrex, ThermoFisher). HPL-derived EVs were added at dilutions of 100,000:1; 10,000:1 and 1000:1 EVs:cell. EGM-2 supplemented with singlequots following the manufacturer’s instructions and EBM-2 basal medium with 4% HSA were used as positive and negative controls, respectively. Samples were incubated for 16 h in a customized live-cell incubation imaging system (Oko Lab, Pozzuoli, Italy) built on an Eclipse *Ti* microscope (Nikon), taking pictures every hour with a 4× objective. Images were processed using NIS software (Nikon) and Photoshop (Adobe, San Jose, CA, USA) and network formation was analyzed using ImageJ software version 1.52p.

### 2.11. Wound Healing Assay

Wound healing surrogate scratch assays were carried out using primary human FBs. In brief, FBs were mitotically inactivated using 1 µg/mL mitomycin C from *S. caespitosus* (Sigma-Aldrich) in complete growth medium (α-MEM supplemented with 10% HPL) for 2.5 h and washed four times with complete growth medium to remove any remaining mitomycin C. FBs were seeded into 2-well culture inserts in 24-well ibiTreat plates (Ibidi, Gräfelfing, Germany) at a density of 4 × 10^4^/cm^2^ and cultured over night for attachment and recovery. The following day, cells were washed once and treated with (i) EVs (TFF and TSEC), (ii) TSEC soluble fractions or (iii) a combination of TSEC-EVs and TSEC soluble fractions (add-back) in α-MEM supplemented with 2% human serum albumin (albunorm^®^, Octapharma GmbH, Vienna, Austria) at an EV-to-cell ratio of 100,000:1. Medium without treatment was used as a negative control, while medium supplemented with 10% HPL served as a positive control. After 4 h of treatment, the silicon inserts were carefully removed to expose the wound (“scratch”) generated by the divider between the wells of each 2-well culture insert. Plates were transferred to a customized Nikon Eclipse Ti inverted microscope equipped with an Okolab on-stage humidified CO_2_ + O_2_-controlled incubator system (Nikon, Vienna, Austria). Cell migration resulting in closure of the scratch between the insert chambers was monitored by taking images with a 10x objective, four fields of view stitched together, every hour, for 24 h. For analysis, images were homogenized and the scratch area was measured using the binary segmentation tool wound area detection in NIS-Elements software. Wound closure was calculated in percentage relative to time point 0 from binary fraction area. For statistical calculations, wound healing results from two individual fibroblast donors and three HPL-EV preparations were analyzed at the 12-h time point.

### 2.12. Statistical Analysis

When comparing multiple groups, a one-way ANOVA with Tukey’s post-test or repeated measures (denoted in the figure legend) were used for statistical analysis. A t-test was used for comparisons between two groups. For grouped data (Figure 1c), a two-way ANOVA repeated measures was employed. All analyses were performed on Graphpad prism software version 8.0.2 (Graphpad Software, San Diego, CA, USA). Statistical significance was considered as follows: **** *p* ≤ 0.0001, *** *p* ≤ 0.001, ** *p* ≤ 0.01, * *p* ≤ 0.05.

## 3. Results

### 3.1. Purification and Characterization of HPL-Derived EVs

We performed three large-scale TFF isolations from 11.10 ± 0.46 L starting volume of media supplemented with 10% HPL [26], and further purified the TFF-EVs by SEC (then termed TSEC-EVs; Figure 1a,b). TFF-EVs presented a high particle count (2.17 ± 0.76 × 10^12^/mL) and high protein concentration (132.30 ± 16.08 mg/mL; all numbers mean ± SD). The SEC purification separated a particle-rich peak of TSEC-EVs (7.20 ± 3.90 × 10^11^/mL; pooling fractions 7–9) and a lipid-protein-rich TSEC soluble fraction peak (protein concentration: 22.00 ± 3.90 mg/mL; pooling fractions 19–21; Figure 1c). TFF-EVs and TSEC-EVs represented a particle recovery of 96.13 ± 3.87% and 48.35 ± 2.05%, respectively. Protein recovery for TFF-EVs was 14.86 ± 6.99% and for TSEC-EVs 0.06 ± 0.003%, confirming depletion of proteins in TSEC-EVs (Figure 1d).

TSEC soluble fractions had a particle recovery of 4.45 ± 2.05% and protein recovery of 4.12 ± 1.59%. We performed western blots to validate the identity of EVs and their respective absence from depleted samples (TFF soluble factors and TSEC soluble fractions). We loaded the samples at the same volume and found both tetraspanins CD9 and CD63 to be abundant in TFF-Evs and, obviously diminished, in TSEC-Evs. Serum components such as has and ApoA1 were present in TFf-EVs, TFF soluble factors and TSEC soluble fractions, and absent in the purified TSEc-EVs (Figure 1e). Thus, TSEC-EV fractions were efficiently depleted of serum lipids and (lipo-) proteins. A faint signal for CD63 in samples of TSEC soluble fractions is compatible with the measured minor particle contamination within the late-running SEC fractions (Figure 1e).

### 3.2. Proteomic Profiling of HPL-Derived Samples

To further characterize HPL-derived samples regarding their protein signature, TMT proteomics were performed using three biological replicates analyzed in three technical replicates. Results showed a complete overlap of technical replicates, and we therefore calculated the average values for each biological replicate. After filtering the proteins detected by removing targets that were only identified by site- and reverse-sequence matches, we obtained a total of 477 proteins. K-means clustering of the recovered proteins revealed a cluster of proteins clearly over-represented in TSEC-EVs (cluster 1), HPL, TFF soluble factors and partly in TSEC-EVs (cluster 2), TSEC-EVs, TSEC-soluble fractions and TFF-EVs (cluster 3), TFF-EVs and TSEC-soluble fractions (cluster 4) as well as HPL and TFF soluble factors (cluster 5; Figure 2a). Principal component (PC) analysis showed that the (I.) TSEC-EV proteome was clearly separated on PC1, representing 64% of the total variance, from the remaining samples, which formed two combined clusters consisting of (II.) HPL and the EV-depleted TFF-derived soluble factors and (III.) TFF-EVs and TSEC-soluble fractions; mostly separated between each other by PC2 (Figure 2b). Of note, the unique clustered (I.) TSEC-EVS showed enrichment for the EV tetraspanin CD9 and the platelet marker integrin beta-3 (ITGB3; CD61) but also for proteins related to the gene ontology (GO) term “receptor-mediated endocytosis” (e.g., integrin beta 1, ITGB1; platelet glycoprotein 4, CD36; ARF gene family members 1 and 6). Proteins related to “lymphocyte mediated immunity” (alpha-2 glycoprotein, AZGP1; hemopexin, HPX) showed a stronger enrichment in the combined clusters (II.) HPL plus TFF sol F. and (III.) TFF-EVs plus TSEC late (Figure 2b). Cluster (III.), containing TFF-EVs plus TSEC-soluble fractions, was enriched for complement factors (C3, C4b, C5 and C1 inhibitor SERPING1) and immunoglobulin A heavy chains (IGHA1 and IGHA2). Interestingly, cluster (III.) also presented increased levels of Apolipoprotein D (Appendix A), which was shown to be an anti-stress molecule induced by oxidative damage and inflammation [40]. Cluster (II.) HPL plus TFF-derived soluble factors also contained complement proteins (C7, C8A and B) and immunoglobulins (IGHG1 and IGHG4) separating these fractions in PC1 from TSEC-EVs. Biological replicate 1 (circles) varied more in relation to the other preparations (identified by triangles and squares), in cluster III. (III.*) more than in TSEC-EV preparations (cluster I.). The overall pattern of PC separation of clusters remained similar across the three biological replicates (Figure 2b). The 20 most upregulated proteins in each HPL-derived sample, together with their corresponding functions, are listed in Table A2.

### 3.3. HPL EVs Support Fibrosphere and Skin-Organoid Formation in the Presence of Platelet Proteins

To address the functionality of HPL-derived samples, in vitro 3D spheroid formation of FBs alone (fibrospheres) or a mixture of KCs, FBs and ECs (assembling into skin-organoids) was used based on previous data [13]. We selected fibrosphere formation as a simplified high throughput readout for 3D spheroid formation based on previous observations with bone marrow stromal cells [41]. In the presence of 2% HPL as a positive control, effective spheroid formation was observed with FBs aggregating into macroscopically visible spheroid structures within 4–6 days. Histology of fibrosphere sections illustrated the compact organization, and immunohistochemistry confirmed maintained positivity for vimentin (Figure 3a–c).

The mixture of KCs + FBs + ECs also organized into spheroid structures in the presence of 2% HPL, reproduced our previous results [13]. After 6 days, skin organoid immunohistochemistry showed a clear surrounding layer of cytokeratin-14 positive KCs and a human vimentin-positive dermal core (Figure 3d–f). HPL-derived TFF-EVs were as potent as 2% HPL at inducing fibrosphere formation (Figure 3g). TSEC-EVs showed a significant loss of function, i.e., significantly reduced spheroid/organoid formation. TSEC-soluble fractions comprising proteins and lipids/lipoproteins were significantly more efficient in inducing spheroid/organoid formation than the separated and ineffective TSEC-EVs. Adding the TSEC-soluble fractions back to their corresponding TSEC-EVs completely reconstituted the fibrosphere and skin-organoid formation. Interestingly, TSEC-soluble fractions (protein-normalized to TFF-EVs) induced fibrosphere and skin-organoid formation, although not to the same extent than the combination of both TSEC-EVs and TSEC-soluble fractions (Figure 3g,h). Because tube-like structures were observed within the dermal core of the skin-organoids (Figure 3f arrowheads), parallel experiments were performed focusing on endothelial cell tube formation [28]. TFF-EVs enhanced in vitro endothelial cell network formation only at highest concentration of 100,000 EVs/cell, whereas TSEC-EVs did not support in vitro angiogenesis at all (Figure A2).

### 3.4. HPL-EVs Inhibit T-Cell Proliferation and Ameliorate Scratch Wound Healing In Vitro in the Presence of Platelet Proteins

Inhibition of PHA- or CD3/28-induced T-cell proliferation was performed to appraise the immunomodulatory capacity of HPL-derived EVs and their corresponding EV-depleted fractions. TFF-EVs were more efficient at inhibiting PHA-induced mitogenesis than in inhibiting CD3/28 crosslinking-induced T cell proliferation. TFF sol. F. and TSEC-EVs were noticeably not effective in inhibiting T-cell proliferation at all (Figure A1 and Figure 4). Surprisingly, a dose-dependent inhibition of T cell proliferation was mediated by TSEC-soluble fractions in both assays. The combination of TSEC-EVs with their corresponding TSEC-soluble fractions (add back) dose-dependently recapitulated the inhibition of T cell proliferation by TFF-EVs, more prominently in PHA- than CD3/28-stimulated T cells (Figure 4).

In a further functional readout, we tested the impact of the different HPL-derived EV and soluble fraction preparations on wound healing in a fibroblast scratch assay in vitro. Scratch wounds in human fibroblast monolayers were mean 60% closed over 24 h in the presence of 10% HPL but did not close in control cultures. TFF-EVs were as efficient. TSEC soluble fractions, presumably containing TFF-EV corona proteins, also significantly augmented scratch wound closure but the protein-depleted TSEC-EVs were not active in this assay. Add-back of TSEC soluble fractions, to their corresponding TSEC-EVs, partly reconstituted wound healing activity (Figure A3).

### 3.5. HPL-EVs Acquire a Protein Corona in a Protein-Rich Environment

Based on these observations, we hypothesized that HPL-EVs can be physically and functionally modified and form a protein corona in a protein-rich environment. To test if protein-poor HPL-derived TSEC-EVs can acquire such a protein corona [24], we first measured zeta potential of TSEC-EVs in the absence or presence of TSEC-soluble fraction proteins. The zeta potential displays the electrical charge of particles in a given medium by their electrophoretic mobility [42]. TSEC-EVs had a significantly more negative charge compared to TFF-EVs and TSEC-soluble fractions. Add-back of TSEC-soluble fractions to TSEC-EVs recapitulated a zeta potential comparable to that of protein-rich TFF-EVs (Figure 5a,b). No statistically significant difference was found when comparing particle size (Figure 5c).

We next used dSTORM super-resolution microscopy to directly question if HPL-derived TSEC-EVs can capture a model protein in a corona-like fashion. TSEC-EVs were therefore marked with a tetramix of AF-647 conjugated antibodies against CD9, CD63 and CD81 in the absence or presence of fluorescent albumin-AF-488 on microchips specifically designed to bind EVs. Albumin-488 alone showed a negligible signal in chips indicating a lack of binding to the EV capturing surface (Figure 6a,d).

The antibody-labeled TSEC-EVs without albumin-AF-488 label showed a homogenously distributed red AF-647 signal (Figure 6b,e). Pre-incubation of TSEC-EVs with fluorescent albumin-AF-488 produced mean 6.75 ± 2.18% double-positive events, i.e., albumin corona-bearing EVs. In addition we found mean 23.13 ± 10.70% single green events (tetramix-negative albumin corona-bearing EVs or albumin aggregates) and mean 70.13 ± 11.62% red events (tetramix-positive EVs lacking albumin; Figure 6c,f). Within total albumin-AF-488 positive events, the mean was 24.03 ± 5.86% co-localized with tetraspanin-positive EVs (Figure 6f). The remaining albumin events that did not co-localize with tetraspanin signals at the EV chip surface might represent tetraspanin-negative EVs or particles or unexpected albumin aggregates that were not observed in the albumin control chips in the absence of EVs.

## 4. Discussion

We performed this study to better understand which of the main fractions of HPL, EVs as compared to the highly potent cytokine and growth factor-rich platelet-derived soluble fraction [5,43], is responsible for HPL’s regenerative and immune modulatory functions. We initially concentrated a total volume of approximately 11 L of 10-fold diluted HPL in a representative cell culture medium down to roughly 60 mL of EVs by TFF. The dilution of HPL in advance of TFF was necessary to reduce the viscosity of HPL before establishing optimized TFF process parameters towards minimum EV loss and high recovery. We opted for cell culture medium to create conditions comparable to stromal cell culture, which led to the discovery of many of the HPL-related regenerative functions, but alternative media may also be used [12,26]. HPL-derived EVs in the TFF retentate were simultaneously separated from platelet-derived soluble factors in the TFF permeate. We confirmed that TFF is a gentle filtration process maintaining the physical properties of EVs at a mean recovery of >90% [24,27,28]. We further purified TFF-EVs by SEC to obtain TSEC-EVs in the early SEC fractions, separated from TSEC-soluble fractions enriched for serum protein components such as HSA and ApoA1. Considering that only three early SEC fractions (fractions 9–11; 6 mL), representing 12% of the total SEC harvest were collected for further TSEC-EV analysis, the recovery of almost 50% of input TFF-EVs indicated reliable localization of TSEC-EVs in the early fractions and efficient processing. EV identity and purity were confirmed according to MISEV2018 guidelines [44] (Table A1).

Clustering of the five HPL-derived preparations analyzed in this study by proteomics followed a clearly organized pattern. Proteins from minimally-processed HPL clustered with TFF soluble factors, indicating that TFF separated a valuable fraction of typical HPL proteins from the TFF-EVs. Various presumably plasma-derived complement factors [45] and immunoglobulins represented functional categories mainly related to immune responses. Interestingly, the co-existence of complement and platelet EVs in HPL and the consecutive fractions [46] does not result in complement-mediated cells lysis as evidenced by superior cell viability and proliferation of multiple human cell types in HPL in vitro, compared to heat-inactivated FBS [47,48]. Proteomic profiles of TFF-EVs and TSEC-soluble fractions also clustered together, indicating a relation between their protein compositions. We hypothesize that at least a proportion of the TSEC-soluble fraction proteins represent the corona protein fraction of the TFF-EVs separated by SEC. Consequently, the unique cluster of TSEC-EVs would represent a corona-depleted HPL-derived EV fraction.

We found that only EVs comprising a lipo-/protein-rich coverage, i.e., TFF-EVs and the re-combination of TSEC-EVs with their soluble fractions, actively contributed to skin organoid formation and immune modulatory function. This led us to hypothesize that the HPL-derived EVs might be physically modified in a protein-rich environment and consequently exert their full function only in the presence of a corona. To support this hypothesis, we performed zeta potential measurements and observed that protein-depleted TSEC-EVs had a more negative surface charge compared to the remaining samples. This is in line with the literature showing that SEC and ultracentrifugation, producing EVs with low protein amounts, give more negative EV charge compared with more ‘gentle’ methods such as precipitation, which also showed higher protein amounts [49]. It was demonstrated that synthetic nanoparticles with known surface charges presented more positive charges when pre-incubated with proteins such as albumin, fibrinogen and γ-globulin [50]. Extracellular vesicles tend to carry a net-negative charge due to the combination of commonly present molecules such as acidic sugars at their surface [42]. One might argue, therefore, that the presence of proteins, especially with a high adsorption rate, can confer increased colloidal stability and positive charges to EVs, as is observed in nanoparticles [50,51].

As a proof of concept, we finally re-established a fluorescent albumin corona on protein-poor TSEC-EVs, reminiscent of protein corona-coating on synthetic nanoparticles or biological EVs when they are exposed to serum proteins [25,50]. We hypothesized that the protein corona is also relevant for EV function when we noticed that stem cell-derived TFF-EVs lost their function when further purified with ultracentrifugation [28], which is a method already shown to deplete the corona of synthetic nanoparticles [50]. Furthermore, our group directly showed that EVs with a reconstituted protein-corona of known growth factors are more potent in inducing angiogenesis as compared to the EVs alone or growth factors alone [24]. In the present work, we reconstituted the HPL EV corona with factors from the same starting material from which the EVs were isolated. Therefore, the native corona of EVs obtained from HPL may be important to exert the full extent of their regenerative potential. Furthermore, using a ‘gentle’ approach for EV isolation, such as TFF, can lead to higher EV potency when compared to methods producing pure, but less functional EVs, such as SEC in the present work.

One additional question that warrants further studies is exactly how stable an EV-protein corona is, depending on the milieu in which they are present. It was shown that a stabilized corona can already be measured by zeta potential at 30 min [50] after incubating synthetic nanoparticles with serum proteins. Because EVs can circulate with access to different tissues and organs [52], it will be interesting to analyze whether the EV-corona remains constant over time in different environments. As a prerequisite for such studies, we already initiated a more detailed analysis of the proteomics results showing clear separation of particularly lipoproteins expected to be involved in metabolic signaling in the different fractions (Figure A4, Appendix A). Lipids and lipoproteins are present in HPL and appear to play a role for HPL and HPL-derived EV function. We found ApoA1 representing a prototypic component of high-density lipoproteins (HDL) to be enriched in western blots of TFF-EVs and TSEC-soluble fractions. ApoA1 presence in TFF-EVs and TSEC-soluble fractions was confirmed in proteomics and could be interpreted as representing HDL contamination [19]. We might speculate that ApoA1 or other lipids or lipoproteins could also contribute to EV corona formation and/or function, as it was already shown on lipid nanoparticles [53]. The experimental design of the current study did not consider an unexpected role of lipids during the assays. A more detailed analysis of lipidomics will be required to better define plasma lipoprotein composition of EV preparations. Ideally, future technology will be able to discriminate EVs from the million-fold more abundant plasma lipoprotein particles [20] directly at the single-particle level.

Components of the immune system may contribute to organ and tissue regeneration after various types of injury [54]. We therefore studied aspects of regeneration by fibrosphere and skin-organoid formation as well as immune modulation in two different T-cell stimulation assays. Accordingly, various types of stromal cells are suggested as putative coronavirus disease 2019 (COVID-19) therapy due to their immunomodulatory and regenerative capacity [55]. In our proteomics study, multiple immune system-related proteins were identified in addition to well-established cytokines and growth factors. These data together with our reproducible high-throughput EV purification strategy build the basis for subsequent more mechanistic studies aiming to define the role of individual EV corona components in different model systems. HPL-derived EVs isolated in a manner permissive to preserve the platelet secretome-derived corona might transmit most potent therapeutic properties.

## 5. Conclusions

We conclude that not only the starting material, but also the isolation/enrichment method(s) of choice can result in EV preparations with different protein coronas, leading to different functional outcomes. We speculate that TSEC soluble fractions in our study comprise TFF-EV-derived corona-derived lipoproteins. Add-back of the TSEC soluble fractions reconstituted TSEC-EV functions argue in favor of this speculation. HPL-derived EVs isolated in a manner permissive to preserve the natural corona might be required to realize the complete therapeutic potential of HPL-EVs and other platelet-derived products.

## Figures and Tables

**Figure 1 biomedicines-10-00238-f001:**
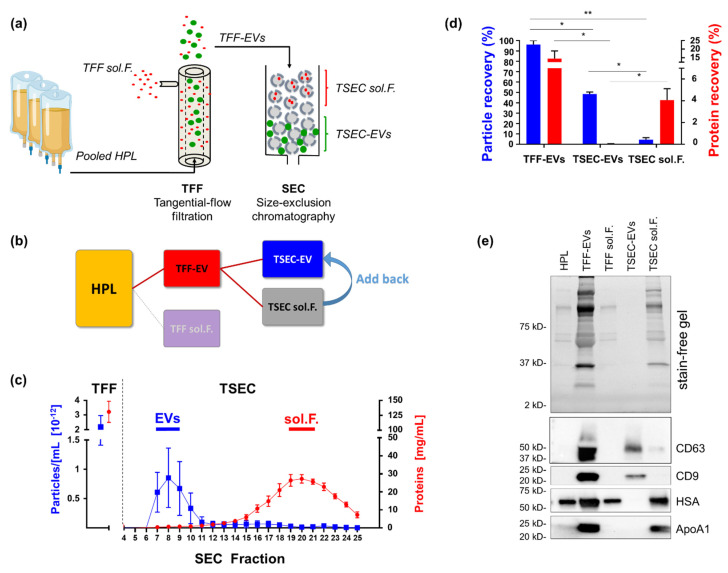
Purification and characterization of human platelet lysate-derived extracellular vesicles (HPL EVs). (**a**) Schematic illustration of EV purification from pooled HPL created using biorender. After two freeze-thaw cycles of expired platelet concentrates and pooling, HPL supernatant was diluted 1:10 in cell culture medium and then separated into soluble factors (sol.F.) and enriched EVs by tangential flow filtration (TFF) [24]. For further enrichment, TFF-EVs were separated by size exclusion chromatography (SEC) into EV (TSEC-EV) and lipid/protein-enriched TSEC-soluble fractions. (**b**) Wire graph illustrating HPL fractionation and the three main fractions analyzed throughout the study. (**c**) Monitoring of protein separation from particles by TFF and subsequent SEC, respectively. Particle concentration measured by tunable resistive pulse sensing (TRPS; blue squares and blue line). Protein concentration analyzed by detergent-compatible (DC) protein assay (red circles and red line). Mean ± SD of three independent experiments performed in triplicate. (**d**) Comparison of particle and protein recovery after EV purification by TFF and TSEC showed particle recovery of 96.13 ± 3.87%, 48.35 ± 2.05% and 4.45 ± 2.05% for TFF-EVs, TSEC-EVs and TSEC-soluble fractions, respectively. Protein recovery was 14.86 ± 6.99%, 0.06 ± 0.003% and 4.12 ± 1.59% for TFF-EVs, TSEC-EVs and TSEC-soluble fractions, respectively. Two-way ANOVA/repeated measures, ** *p* ≤ 0.01, * *p* ≤ 0.05. (**e**) Western blot for tetraspanins CD63 and CD9, human serum albumin (HSA) and apolipoprotein A1 (ApoA1). Representative depiction, two independent western blot experiments were performed.

**Figure 2 biomedicines-10-00238-f002:**
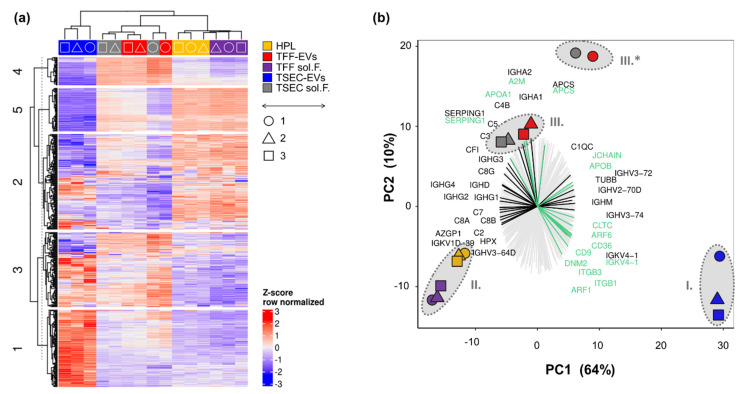
Proteomic profiling of HPL-derived samples. (**a**) Tandem mass-tag (TMT) proteomics of HPL, TFF-EVs, TFF sol.F., TSEC-EVs and TSEC-soluble fractions were analyzed by unbiased clustering in a heatmap indicating five main clusters (1–5, left side). Protein detection levels were row-wise Z-score normalized. Color code as indicated with most replicates grouping together (top) with symbols indicating the three independent experiments (◯△☐). (**b**) Bi-plot including a principal component (PC) analysis and a protein recovery loading plot. Three major clusters composed of (**I.**) TSEC-EVs (blue symbols), (**II.**) HPL + TFF sol.F. (yellow and violet symbols) and (**III.**) TFF-EVs + TSEC-soluble fractions (red and grey symbols, respectively). (III*) Biological replicate 1 differed more in relation to other samples, but was still included within cluster III. Three biological replicates, each performed in three technical replicate runs. A loading plot is also showing the contribution of the 477 detected proteins towards the different clusters. Lines pointing towards a cluster show an enrichment of proteins in this specific cluster. We highlighted in black the “lymphocyte mediated immunity” related proteins and in green the “receptor-mediated endocytosis” proteins.

**Figure 3 biomedicines-10-00238-f003:**
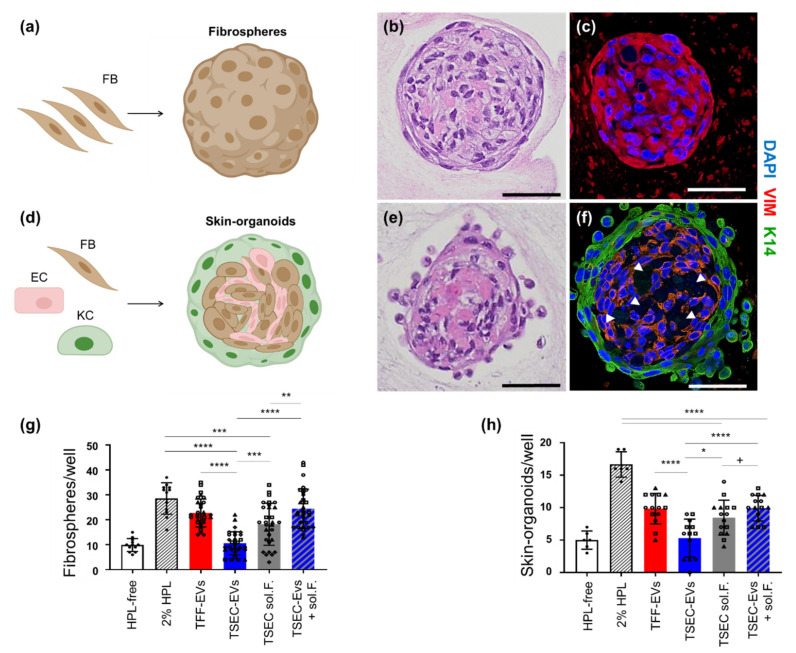
TSEC-EVs enhance fibrosphere and skin-organoid formation in the presence of platelet proteins. (**a**–**c**) Fibrosphere formation. (**a**) Cartoon illustrating monotypic fibroblast (FB) aggregation. (**b**) Hematoxylin eosin staining and, (**c**) immunostaining of representative fibrospheres showing compact and human vimentin (VIM)-positive structures. (**d**–**f**) Skin-organoid formation. (**d**) Cartoon illustrating aggregation of primary human FBs, keratinocytes (KCs) and endothelial cells (ECs) when seeded in a 2:1:1 ratio in a permissive environment. (**e**) Hematoxylin and eosin staining of a representative skin organoid presenting a compact core with stratified envelope. (**f**) Immunohistochemistry confirmed a human VIM-positive dermal core with vascular-like empty spaces (white arrow heads) and a surrounding cytokeratin K14-positive KC layer. Scale bars: 50 µm. Schematic cartoons created using Biorender. (**g**) Quantification of fibrospheres per well was performed after incubation of 10^5^ EVs/cell. Matching protein concentrations of TSEC-soluble fractions 19–21 and TSEC-EVs plus their corresponding TSEC-soluble fractions (TSEC-EVs + soluble fractions) after 6 days with 5000 FBs per well (*n* = 5). (**h**) 2500 KCs, 1250 FBs and 1250 ECs per well for skin-organoid quantification were analyzed accordingly (*n* = 3). Each symbol (circles, squares and triangles) represents a different biological replicate (*n* = 5 for fibrosphere assay, *n* = 3 for skin-organoid assay), each performed in hexaplicates. One-way ANOVA/Tukey, **** *p* ≤ 0.0001, *** *p* ≤ 0.001, ** *p* ≤ 0.01, * *p* ≤ 0.05; **^+^**
*p* ≤ 0.05; one-tailed *t*-test.

**Figure 4 biomedicines-10-00238-f004:**
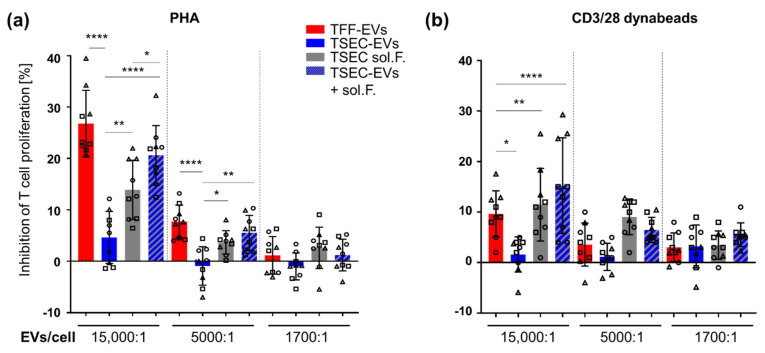
TSEC-EVs inhibit T-cell proliferation in the presence of platelet proteins. Peripheral blood mononuclear cells (PBMCs) containing app. 80% T cells [37,38] were incubated with different concentrations of TFF-EVs, TSEC-EVs, matching TSEC-soluble fractions or the combination of both (TSEC-EVs + soluble fraction add-back) for 4 days. (**a**) Proliferation of T cells was induced by phytohemagglutinin (PHA) and inhibition of T cell proliferation was most effective in the presence of TFF-EVs and TSEC-EVs + soluble fraction add-back. TSEC-EVs showed virtually no effect and TSEC-soluble fractions had a partial inhibitory effect. (**b**) After CD3/28 stimulation, a significant loss of function was observed comparing TSEC-EVs to TFF-EVs for their capacity to inhibit T cell proliferation. No significant difference was found between TSEC-soluble fractions and TSEC-EVs + soluble fractions add-back, both inhibiting T cell proliferation. One-way ANOVA, repeated measures, **** *p* ≤ 0.0001, ** *p* ≤ 0.01, * *p* ≤ 0.05. Each symbol (circles, squares and triangles) represents a different biological replicate (*n* = 3) performed in triplicate.

**Figure 5 biomedicines-10-00238-f005:**
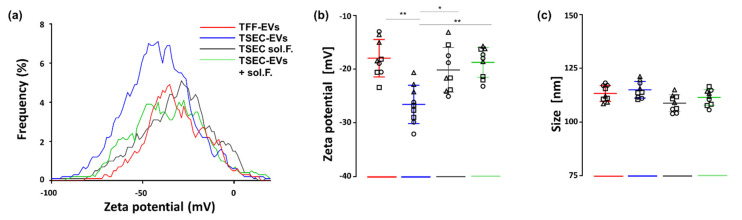
TSEC-EV zeta potential in the absence or presence of a protein-rich environment. (**a**) Zeta-potential frequency distribution (depicted as rolling average) and (**b**) voltage was measured with a ZetaView instrument by evaluating the particles’ migration in a pressure-free environment, strictly by charge. HPL-derived TFF-EVs, TSEC-EVs alone, and TSEC-EVs after add-back of their corresponding TSEC-soluble fractions (TSEC-EVs + soluble fractions) are shown. (**c**) There was no statistically significant difference in particle size. One-way ANOVA, repeated measures, ** *p* ≤ 0.01, * *p* ≤ 0.05. Each symbol (circles, squares and triangles) represents a different biological replicate (*n* = 3) tested in triplicate.

**Figure 6 biomedicines-10-00238-f006:**
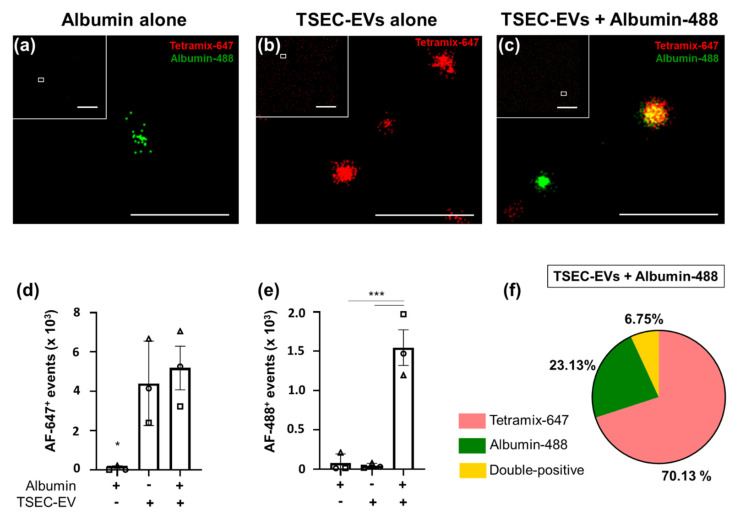
Artificial reconstitution of a protein corona on HPL EVs. TSEC-EVs (5.4 × 10^8^) were labeled with anti-CD9, CD63 and CD81 tetramix-AF-647 with or without prior albumin-AF-488 corona formation and loaded onto EV profiler chips (ONI). High resolution pictures with overview inserts are shown. (**a**) Negative control albumin-AF-488 solution in the absence of EVs did not bind to the chip; one single positive presumably albumin aggregate is shown in magnification. (**b**) Control tetramix-AF-647-labeled TSEC-EVs without albumin label bound to the chip showing red signal, with negligible signal in the green channel. (**c**) Albumin-AF-488 pre-labeled and tetramix-AF-647 stained TSEC-EV samples produced easily detectable double-positive signals. (**d**) Quantification of AF-647 signals and (**e**) quantification of AF-488 signals on chips loaded as indicated with albumin solution or TSEC-EVs without or with previous fluorescent albumin corona formation. (**f**) In chips loaded with albumin-AF-488-prelabeled TSEC-EVs, represented in (**c**), 6.75 ± 2.18% double-positive, 70.13 ± 11.62% tetramix-647 positive and 23.13 ± 10.70% AF-488 positive events were detected. One-way ANOVA, Tukey, *** *p* ≤ 0.001, * *p* ≤ 0.05. Each symbol (circles, squares and triangles) represents a separate independent experiment (*n* = 3). Scale bars: 10 µm (low resolution inserts) and 0.5 µm (main panels **a**–**c**).

## Data Availability

Appendix A consists of the normalized log2 fold-changes for all proteins found on tandem-mass tag proteomics. Information concerning EV isolation and characterization has also been uploaded on EV-TRACK (EV-TRACK ID EV210381).

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
