# Peer review of "Synergy of Human Platelet-Derived Extracellular Vesicles with Secretome Proteins Promotes Regenerative Functions"

_biomedicines, 2022, doi:10.3390/biomedicines10020238_

Round 1

Reviewer 1 Report

Summary:

The goal of this study was to investigate the characteristics of extracellular vesicles (EVs) isolated from human plasma and their regenerative capabilities. The results of this study emphasize the importance of isolation techniques of EVs to mediate their biological functions. This study as also highlights the importance of the corona and the multitude of roles it plays within the functioning and uptake of EVs. Overall, this paper is well constructed and follows a logical experimental scheme.

Comments:
1. Although the goal of this study was to investigate the regenerative ability of exosomes isolated from human plasma lysate, proteomics revealed that exosomes and exosomal corona proteins represent completely unique clusters from the human protein lysate while the soluble factors actually cluster with human protein lysate. Why were soluble factors never investigated or addressed in later studies?

  1. Figure 6f: Needs clarification. The figure shows tetramix only (70.13%), albumin only (23.13%), and double positive (6.75%). However, these numbers do not match in the description of this figure. Why an Albumin-488 group only was considered in calculations? What is the importance of Albumin-488?

Author Response

Dear Vesta, dear reviewers:

Thank you for the thorough review of our manuscript. We appreciate the propositions provided, we performed additional experiments as requested and we addressed all other questions point by point below. All changes within the manuscript were done in ‘track mode’. Please find our response to the reviewer requests below:

Reviewer 1:

  1. Although the goal of this study was to investigate the regenerative ability of exosomes isolated from human plasma lysate, proteomics revealed that exosomes and exosomal corona proteins represent completely unique clusters from the human protein lysate while the soluble factors actually cluster with human protein lysate. Why were soluble factors never investigated or addressed in later studies?

Thank you for making this point. Indeed, we evaluated TFF soluble factors in preliminary immune modulation assay and found no effect. Therefore, we only continued with the TFF EVs for remaining assays. This information is now included in the revised manuscript text and in a new figure A1. We also state now more precisely, that “…protein-rich TFF-EVs or a combination of the protein-depleted TSEC-EVs with add-back of protein-rich TSEC-soluble fractions were efficient in four surrogate readouts for regenerative function. The EV-depleted TFF soluble fractions and the protein-depleted TSEC-EVs were not active in these assays”. A wire graph illustrating the EV separation tree was included as requested by reviewer #3.

  1. Figure 6f: Needs clarification. The figure shows tetramix only (70.13%), albumin only (23.13%), and double positive (6.75%). However, these numbers do not match in the description of this figure. Why an Albumin-488 group only was considered in calculations? What is the importance of Albumin-488?

Our fault – thank you for identifying the error. The numbers on the figure and figure legend were indeed not matching. We corrected the mistake. Thank you also for indicating possible misinterpretation of figure 6f. We improved figure legend and results section accordingly.  

Reviewer 2 Report

Platelet-rich plasma is a promising regenerative therapeutic with controversial efficacy.

This study analyzes   not only the starting material, but also the isolation/enrichment method(s) of choice can result in EV preparations with different protein coronas, leading  to different functional outcomes. HPL-derived EVs isolated in a manner permissive to preserve the natural corona might be required to realize the complete therapeutic potential of HPL-EVs and other platelet-derived products.

The paper is well organized

  •    Summary  includes the main points
  •   In the introduction section, it provides sufficient information on specific background information on the topic. The results is Ok 
  • The discussion section contains information that develops and supports the thesis
  • The figures are very clear and help to understand the text

Author Response

Reviewer 2:

The paper is well organized; Summary  includes the main points;   In the introduction section, it provides sufficient information on specific background information on the topic. The results is Ok .The discussion section contains information that develops and supports the thesis. The figures are very clear and help to understand the text

Thank you for the positive comments.

Reviewer 3 Report

In this manuscript, authors attempted to characterize the platelet-derived factor responsible for tissue regeneration. As mentioned in this manuscript, there has been a clinical interest in platelet containing fractions in tissue healing yet, due to inconsistency in results, platelet use in clinical application are limited. In this manuscript, authors identified that platelet fraction promotes formation of fibrospeheres and organoids and inhibits lymphocyte (T cell) proliferation. Next, authors made series of fractions using tangential flow filtration (TFF) and size-exclusion chromatography (SEC) and identified that, soluble fraction of SEC retained the platelet functions but not the SEC EVs fraction. Moreover, there is an added of soluble fraction of SEC if combined with SEC EVs to fully recapitulate the platelet functions, lack of tissue regeneration capacity for EVs is quite interesting! Authors also performed proteomics on various platelet factions. Overall, this is an interesting work in the field of tissue regeneration. However, there are several comments (listed below).

  1. While Figure 1 explains the experimental complexity and generation of different platelet fractions, from reading the abstract alone it is difficult for the reader to follow the findings (especially with the TSEC-EVs and TSEC-late fractions). Instead of “TSEC-late fractions,” authors may choose a term line EV-free or soluble fraction. A simplified wire diagram or re-writing the abstract is required.
  2. Again, a wire diagram for 2.2 (under Materials and Methods) is necessary to reduce the complexity.
  3. In 3.1, authors again provided methodology while explaining the results. This approach reduces reader’s enthusiasm. Authors should focus on the results here.
  4. In Figure 2, authors performed proteomics work but there is no follow-up in the subsequent figures. While a follow-up study is expected, it is not always required. However, authors should move this Figure 2 to the last figure. Because there will be a tremendous in interest to learn from this proteomics work, authors should elaborate the functional role of each protein in a separate table.
  5. Current study focused on fibrospehere and organoid formation and T cell proliferation assays. However, there are other assays which are well-recognized classic experiments to understand epithelial functions (eg. wound healing assay). Do authors have data on such additional end points?
  6. Based on initial question of platelet fraction responsible for tissue regeneration, Figure 5 is less interesting and can be moved to supplement. Figure 6 can be new Figure 4 and Figure 2 should be new Figure 5.  

Minor:

  1. Authors performed proteomics but there is no mention in the abstract!

Author Response

Reviewer 3:

  1. While Figure 1 explains the experimental complexity and generation of different platelet fractions, from reading the abstract alone it is difficult for the reader to follow the findings (especially with the TSEC-EVs and TSEC-late fractions). Instead of “TSEC-late fractions,” authors may choose a term line EV-free or soluble fraction. A simplified wire diagram or re-writing the abstract is required.

Thank you for raising these points. Terminology was changed to ‘TSEC soluble fractions’ as suggested. A simplified wire diagram was added in figure 1b as requested. The abstract was improved.

  1. Again, a wire diagram for 2.2 (under Materials and Methods) is necessary to reduce the complexity.

As requested by this reviewer, we now refer to the wire diagram in figure 1 at the end of 2.2.

  1. In 3.1, authors again provided methodology while explaining the results. This approach reduces reader’s enthusiasm. Authors should focus on the results here.

Thank you. Methods details moved to section 2.2 as requested.

  1. In Figure 2, authors performed proteomics work but there is no follow-up in the subsequent figures. While a follow-up study is expected, it is not always required. However, authors should move this Figure 2 to the last figure. Because there will be a tremendous in interest to learn from this proteomics work, authors should elaborate the functional role of each protein in a separate table.

As requested by this reviewer we performed additional bioinformatics analysis and provide a supplementary table 2 containing the top-20 differentially expressed proteins recovered by mass-tag proteomics of the different subfractions analyzed, as well as an additional pathway analysis (new figure A4). Another table summarizing the complete proteomics details is provided as supplementary file. We kept the proteomics data as figure to avoid excessive re-writing of the results and discussion parts in the limited time available and rather opted for performing the interesting wound healing experiments suggested by this reviewer (below).

  1. Current study focused on fibrospehere and organoid formation and T cell proliferation assays. However, there are other assays which are well-recognized classic experiments to understand epithelial functions (eg. wound healing assay). Do authors have data on such additional end points?

We performed additional wound healing experiments as suggested using fibroblasts as the mediators of wound bed closure, in addition. We found significant enhancement of wound closure by TFF-EVs and TSEC sol.F. but not TSEC-EVs. HPL served as a positive control. These results are now described and shown in an additional figure A3.

  1. Based on initial question of platelet fraction responsible for tissue regeneration, Figure 5 is less interesting and can be moved to supplement. Figure 6 can be new Figure 4 and Figure 2 should be new Figure 5.

We consider the zeta potential as key results of enormous interest for nanoscience and EV experts and therefore opted against moving this figure to the supplement but appreciate the comment.

Minor:

  1. Authors performed proteomics but there is no mention in the abstract!

Thank you – corrected accordingly.

We hope that all aspects of the minor revision are sufficiently addressed.

Yours sincerely

D. Strunk & team